# Supraclavicular brown adipocytes originate from *Tbx1+* myoprogenitors

Zan Huang[1,2,3◐], Chenxin Gu[3◐], Zengdi Zhang[3], Rini Arianti[4], Aneesh Swaminathan[3], Kevin Tran[3], Alex Battist[3], Endre Kristóf[4], Hai-Bin Ruan[3,5]*

**1** Laboratory of Gastrointestinal Microbiology, Jiangsu Key Laboratory of Gastrointestinal Nutrition and Animal Health, College of Animal Science and Technology, Nanjing Agricultural University, Nanjing, China, **2** National Center for International Research on Animal Gut Nutrition, Nanjing Agricultural University, Nanjing, China, **3** Department of Integrative Biology and Physiology, University of Minnesota Medical School, Minneapolis, Minnesota, United States of America, **4** Department of Biochemistry and Molecular Biology, Faculty of Medicine, University of Debrecen, Debrecen, Hungary, **5** Center for Immunology, University of Minnesota Medical School, Minneapolis, Minnesota, United States of America

◐ These authors contributed equally to this work.
* hruan@umn.edu

## Abstract

Brown adipose tissue (BAT) dissipates energy as heat, contributing to temperature control, energy expenditure, and systemic homeostasis. In adult humans, BAT mainly exists in supraclavicular areas and its prevalence is associated with cardiometabolic health. However, the developmental origin of supraclavicular BAT remains unknown. Here, using genetic cell marking in mice, we demonstrate that supraclavicular brown adipocytes do not develop from the *Pax3+*/*Myf5+* epaxial dermomyotome that gives rise to interscapular BAT (iBAT). Instead, the *Tbx1+* lineage that specifies the pharyngeal mesoderm marks the majority of supraclavicular brown adipocytes. *Tbx1Cre*-mediated ablation of peroxisome proliferator-activated receptor gamma (PPARγ) or PR/SET Domain 16 (PRDM16), components of the transcriptional complex for brown fat determination, leads to supraclavicular BAT paucity or dysfunction, thus rendering mice more sensitive to cold exposure. Moreover, human deep neck BAT expresses higher levels of the *TBX1* gene than subcutaneous neck white adipocytes. Taken together, our observations reveal location-specific developmental origins of BAT depots and call attention to *Tbx1+* lineage cells when investigating human relevant supraclavicular BAT.

## Introduction

Brown adipose tissue (BAT) is a thermogenic organ found in almost all mammals that dissipates energy as heat, thus contributing to homeostatic regulation of body temperature and metabolic physiology. In newborn humans, the predominant BAT depot is located in the interscapular region (iBAT). Through poorly understood mechanisms [1], iBAT undergoes progressive involution, scatters around the back during adolescence, and becomes undetectable in most adults [2–4]. In adult humans, metabolically active BAT instead exists in cervical and supraclavicular areas, collectively referred as neck BAT [5–9]. The

**Data Availability Statement:** All relevant data are within the paper and its Supporting information files.

**Funding:** This work was supported by the National Natural Science Foundation of China, China (32170847) to Z.H., National Research, Development and Innovation Office of Hungary (NKFIH-FK131424) to E.K., and Department of Integrative Biology and Physiology Grant Accelerator Program to H.-B.R. The funders had no role in study design, data collection and analysis, decision to publish, or preparation of the manuscript.

**Competing interests:** The authors have declared that no competing interests exist.

**Abbreviations:** BAT, brown adipose tissue; CPM, cardiopharyngeal mesoderm; HFD, high-fat diet; iBAT, interscapular brown adipose tissue; LPS, lipopolysaccharide; PPARγ, peroxisome proliferator-activated receptor gamma; scBAT, supraclavicular brown adipose tissue; SVF, stromal vascular fraction.

prevalence of neck BAT is inversely associated with body mass index [10,11] and declines as a function of age [9,12,13], indicating the potential involvement of neck BAT dysfunction in the development of obesity and related metabolic disorders [14]. However, the lineage origins and mechanisms for age-dependent functional decline of neck BAT remain almost unknown.

In the past decade, tremendous efforts have been made in our understanding of the development, recruitment, and activation of BAT. Nonetheless, most mechanistic studies were performed on rodent iBAT, due to its large size and easy accessibility. BAT and skeletal muscle have shared metabolic features and embryonic origins. Genetic fate mapping experiments in mice demonstrate that the dermomyotome regions of the somites, marked by the expression of transcription factors including *Pax3*, *Pax7*, *Meox1*, and *Myf5*, give rise to most fat cells within the interscapular and retroperitoneal adipose depots [15–20]. The fact that these lineages trace to dorsal-anterior-located muscle, brown and white adipocytes suggests that they are location markers, rather than identity markers. Therefore, it is unlikely, although not tested or reported, that *Pax3*+/*Myf5*+ myoprogenitors form brown adipocytes in ventral neck BAT that has a very distinct location compared to dorsal-anterior BAT.

In vertebrates, head and neck muscles arise from the unsegmented cranial mesoderm, in distinction to somite-derived trunk muscles [21]. Transcriptional factors such as *Tbx1*, *Ptx2*, and *Islet1* specify the cardiopharyngeal mesoderm (CPM) that gives rise to muscles of the head and heart [22–24]. Supraclavicular BAT (scBAT) in mice is located in a region analogous to human neck BAT (Fig 1A). Though smaller than iBAT, subscapular and supraspinal (also termed as posterior cervical) BAT depots in the dorsal trunk (Fig 1B), scBAT possesses similar thermogenic activity and regulation [25,26]. However, the developmental origins of scBAT adipocytes have not been defined. We hypothesized that the CPM also contributes to connective tissues in the neck region, including scBAT. In this study, taking advantage *Pax3*^*Cre*, *Myf5*^*Cre*, and *Tbx1*^*Cre*-mediated lineage tracing and gene ablation, we identified the location-specific myogenic progenitors for scBAT versus iBAT in mice. Importantly, scBAT as *Tbx1*-progeny appears to be true in humans as well. This knowledge can be leveraged in the future to investigate location-dependent functions of BAT and to target scBAT specifically for metabolic improvements.

## Results

### *Pax3*+ progenitors rarely give rise to supraclavicular brown adipocytes

*Pax3*, together with its orthologue *Pax7*, initiate a transcriptional cascade including *Myf5* and *Myod* for myogenesis. To determine if supraclavicular BAT arises from *Pax3*+ myogenic progenitors, we mated *Pax3*^*Cre* to *Rosa26*^*LSL-mT/mG* reporter mice (Fig 1C). *Pax3*-derived cells express membrane-tethered GFP (mG), while those non-*Pax3* progeny cells express membrane-tethered tdTomato (mT). As expected, brown adipocytes within dorsal BAT including interscapular, subscapular, and supraspinal depots were almost exclusively GFP+ (Fig 1D and 1E), indicating their *Pax3*-lineage origin. Mouse scBAT localizes in the ventral site of the neck, beneath the submandibular gland and tightly connected to the jugular vein (Fig 1F). We dissected both medial and lateral scBAT depots, which are above and below the jugular vein respectively. Very rare brown adipocytes in these scBAT depots (approximately 6.7% in medial and 0.3% in lateral) were GFP+ (Fig 1G and 1H). No adipocytes in either iBAT or scBAT were labeled in Cre-negative animals (S1 Fig), validating reagents and methods used for lineage tracing in the study. These data demonstrate that most scBAT adipocytes are not progeny of somite myogenic progenitors.

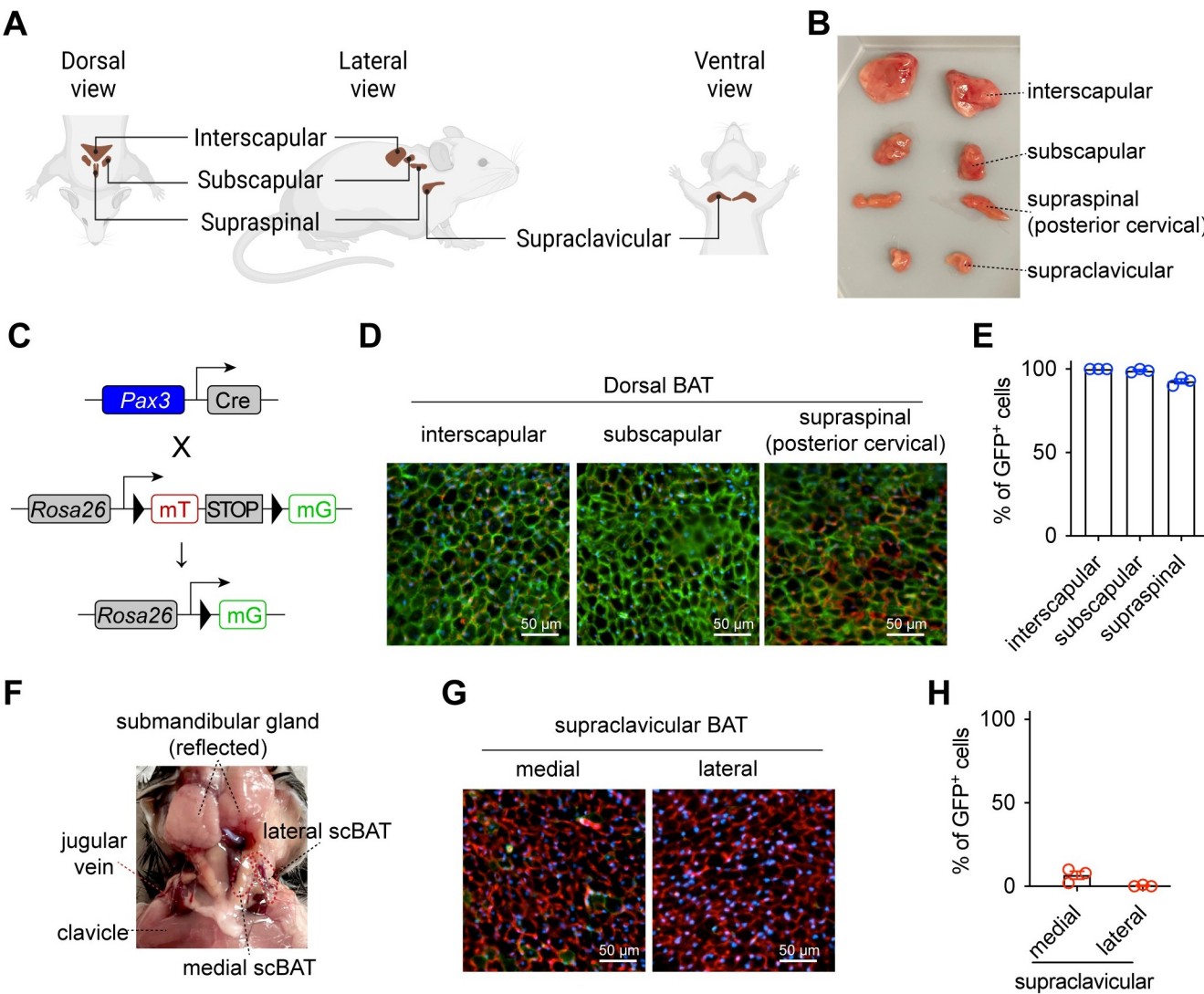

**Fig 1. scBAT does not arise from *Pax3*⁺ progenitor cells. (A)** Schematic representation of the location of peri-scapular and neck BAT in mice. **(B)** Representative photo of major BAT depots examined in this study. **(C)** Generation of the reporter mice for *Pax3*⁺ cells. **(D)** Fluorescent images of dorsal BAT depots from 5-week-old male *Pax3-mTmG* mice (green = mG, red = mT, blue = DAPI, scale = 50 μm). **(E)** Frequency of GFP⁺ cells within each BAT depot (*n* = 3). **(F)** Anatomic location of supraclavicular BAT depot (indicated with red dotted line). **(G)** Fluorescent images of medial and lateral scBAT from *Pax3-mTmG* mice. (green = mG, red = mT, blue = DAPI, scale = 50 μm). **(H)** Quantification of GFP⁺ cells as a percentage of total adipocytes (*n* = 3). Data are presented as mean ± SEM. Extended data are listed in S1 Data. BAT, brown adipose tissue; scBAT, supraclavicular brown adipose tissue.

### *Myf5*⁺ progenitors seldom give rise to scBAT adipocytes

*Myf5*⁺ myogenic progenitors contribute to trunk and limb muscles, and dorsal adipose tissues including iBAT. To determine if scBAT arises from *Myf5*⁺ cells, we mated *Myf5^Cre^* to *Rosa26^LSL-mT/mG^* reporter mice (Fig 2A). Similar to the *Pax3^Cre^* reporter, adipocytes within dorsal BAT depots were mostly mG⁺ (Fig 2B and 2D), representing their somite origins. However, only approximately 7% of adipocytes in the medial scBAT were GFP-labeled and essentially no adipocytes were labeled in the lateral scBAT (Fig 2C and 2D). Collectively, our *Pax3^Cre^* and *Myf5^Cre^* cell marking data demonstrate that scBAT and iBAT do not share the same myogenic origins.

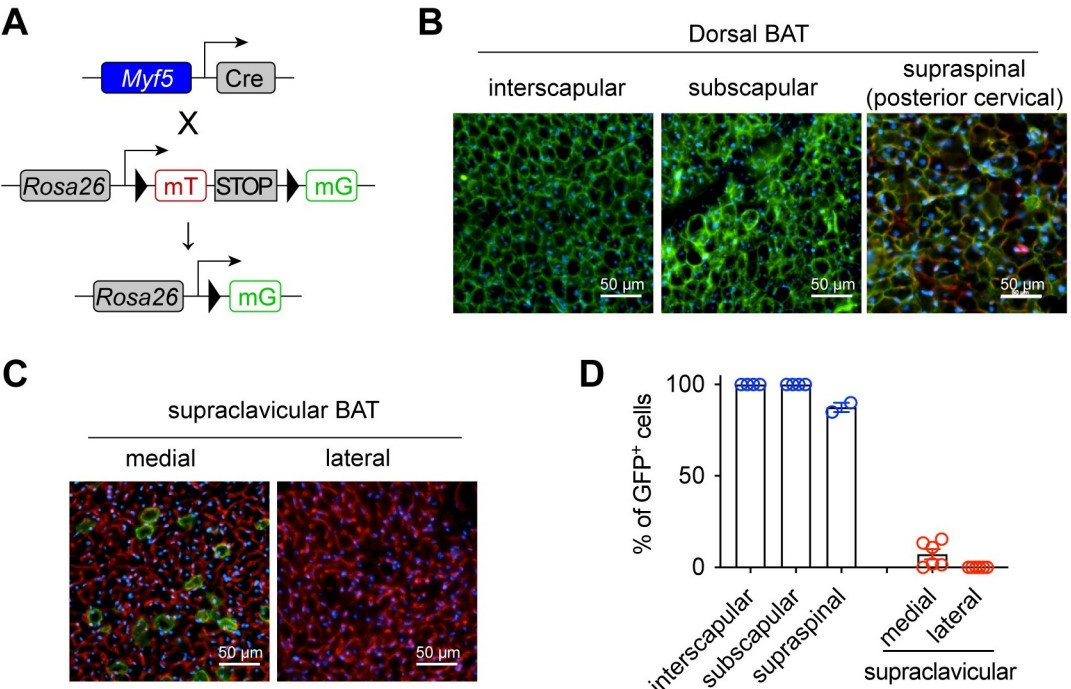

**Fig 2. scBAT does not arise from *Myf5*⁺ precursor cells. (A)** Generation of the reporter mice for the *Myf5*⁺ lineage cells. **(B, C)** Fluorescent images of dorsal BAT (B) or scBAT (C) from 2-month-old female *Myf5-mTmG* mice (green = mG, red = mT, blue = DAPI, scale = 50 μm). **(D)** Frequency of GFP⁺ cells within each BAT depot (*n* = 2–5). Data are presented as mean ± SEM. Extended data are listed in S1 Data. BAT, brown adipose tissue; scBAT, supraclavicular brown adipose tissue.

To validate the lineage marking data, we then generated *Pparg* knockout mice specifically in *Myf5*⁺ cells (*Pparg^ΔMyf5^*). The *Pparg* gene encodes the master transcriptional factor for adipogenesis—peroxisome proliferator-activated receptor gamma (PPARγ). As a result, severe BAT paucity was observed in the interscapular and subscapular depots (Fig 3A and 3B). Less obvious mass reduction was seen in the supraspinal BAT (Fig 3A and 3B), possibly due to the existence (approximately 15%) of non-*Myf5*⁺ progeny cells in this depot (Fig 2B and 2D). Western blotting showed a complete loss of PPARγ protein and significant reduction in UCP1 expression (Fig 3C). In contrast, scBAT did not reduce its size in *Pparg^ΔMyf5^* mice (Fig 3D). Instead, there was a trending increase in weight when compared to littermates (Fig 3E), suggesting a potential compensation for iBAT paucity. *Myf5^Cre^* does not target scBAT, thus no change in PPARγ or UCP1 expression was observed (Fig 3F). Taken together, *Myf5*⁺ myogenic progenitors are essential for the development of dorsal-located BAT, but not ventral-located scBAT.

## scBAT adipocytes arise from *Tbx1*⁺ progenitors

The *Tbx1* gene is expressed in CPM that gives rise to the branchiomeric and transition zone muscles between head and trunk. To determine if *Tbx1*⁺ progenitors mark the nearby scBAT, we generated *Tbx1^Cre^*-dependent mT/mG reporter mice (Fig 4A). The tracing of the CPM was confirmed by the mT-labeling of scapular muscle (*Myf5*-lineage) and the mG-labeling of clavicular muscle (*Tbx1*-lineage) (S2 Fig). In consistent with their origin as *Myf5*⁺ progeny, brown adipocytes in dorsal BAT, including interscapular, subscapular, and supraspinal depots, were not labeled at all by mG in *Tbx1-mT/mG* mice (Fig 4B and 4D). In contrast, nearly 50% of scBAT adipocytes are mG⁺ (Fig 4C and 4D). We saw the even distribution of *Tbx1*-lineage

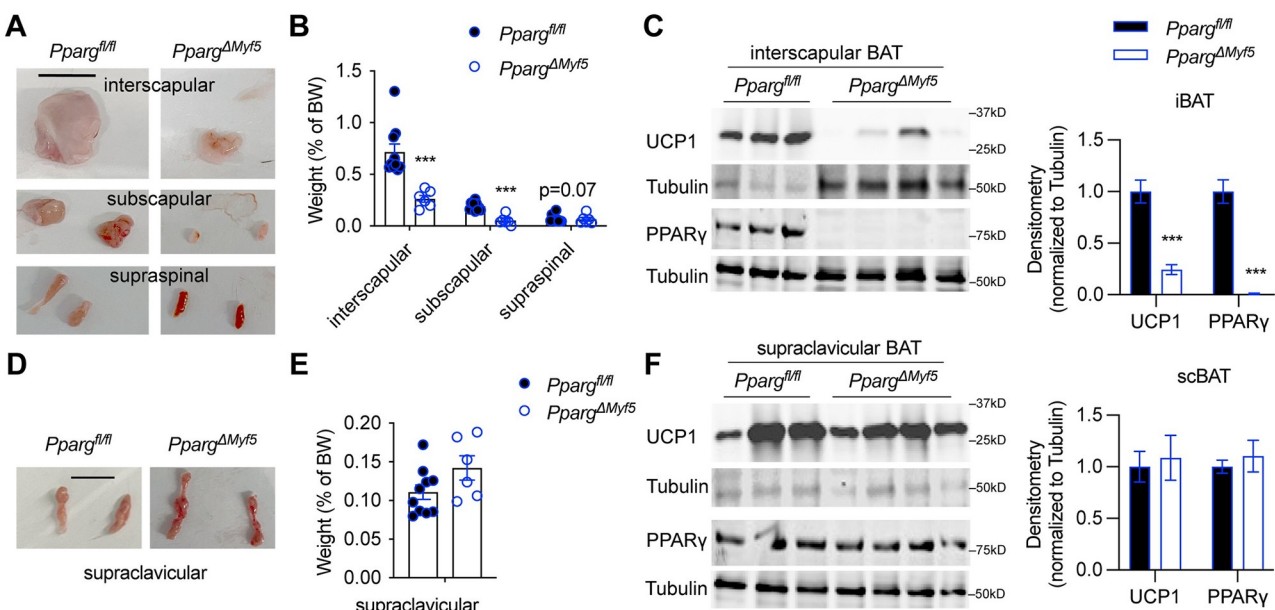

**Fig 3. Intact scBAT in mice with PPARγ deficiency in the *Myf5*+ lineage. (A, B)** Indicated dorsal BAT depots from 8-month-old *Pparg^{f/f}* (mixed sex, *n* = 10) and *Pparg^{ΔMyf5}* mice (mixed sex, *n* = 6) were isolated, photographed (A, scale = 1 cm), and weighed (B). **(C)** Representative immunoblotting of UCP1 and PPARγ with iBAT proteins from *Pparg^{f/f}* and *Pparg^{ΔMyf5}* mice. Densitometry (*n* = 9–10) is shown to the right. **(D, E)** scBAT depots were isolated from 8-month-old *Pparg^{f/f}* (*n* = 10) and *Pparg^{ΔMyf5}* mice (*n* = 6), photographed (D, scale = 1 cm), and weighed (E). **(F)** Representative UCP1 and PPARγ expression in scBAT from *Pparg^{f/f}* and *Pparg^{ΔMyf5}* mice. Densitometry (*n* = 9–10) is shown to the right. Data are presented as mean ± SEM. ***, *p* < 0.01 by unpaired Student's *t* test. Extended data are listed in S1 Data. BAT, brown adipose tissue; iBAT, interscapular brown adipose tissue; PPARγ, peroxisome proliferator-activated receptor gamma; scBAT, supraclavicular brown adipose tissue.

adipocytes across the whole scBAT, which is preserved in aged mice (Fig 4E). Salivary gland can be found to be connected with the medial end of scBAT. Salivary gland is derived from oral epithelium, thus not labeled by mG (Fig 4E). It is currently unclear what the identity of these mT+ non-*Tbx1* progeny adipocytes in the scBAT depot.

## scBAT contributes to temperature maintenance in female mice

To evaluate the functional contribution of scBAT to systemic metabolism, we generated *Tbx1*-specific *Pparg* knockout mice (*Pparg^{ΔTbx1}*). In female *Pparg^{ΔTbx1}* mice, PPARγ deficiency leads to a specific decrease of scBAT weight (Fig 5A), but not of any dorsal depots including iBAT (Fig 5B–5D). The approximately 50% reduction in scBAT weight is consistent with the labeling efficiency of *Tbx1*+ progenitors (Fig 4) and the knockout efficiency of the *Pparg* genes in *Pparg^{ΔTbx1}* scBAT (S3A Fig). Body mass and weights of WAT and skeletal muscle were not affected in these *Pparg^{ΔTbx1}* female (S3B and S3C Fig). Areas indicating adipose "whitening" and tissue degeneration could be found in *Pparg^{ΔTbx1}* scBAT (Fig 5E). RT-qPCR revealed a significant down-regulation of total transcripts of thermogenic genes in scBAT but not iBAT (Fig 5F and 5G). As a result, female *Pparg^{ΔTbx1}* mice reduced more body temperature when challenged with cold, compared to *Pparg^{f/f}* controls (Fig 5G). Similar scBAT paucity but intact dorsal BAT depots were observed in male *Pparg^{ΔTbx1}* mice (Fig 5I–5L). Body, WAT, and muscle weights were comparable between 2 genotypes in males (S3D and S3E). Histological assessment of scBAT found more unilocular adipocytes in *Pparg^{ΔTbx1}* mice (Fig 5M), indicative of the loss of brown identity in PPARγ-deficient adipocytes. Some thermogenic genes like *Prdm16* and *Dio2* were specifically down-regulated in male scBAT (Fig 5N), but not in iBAT

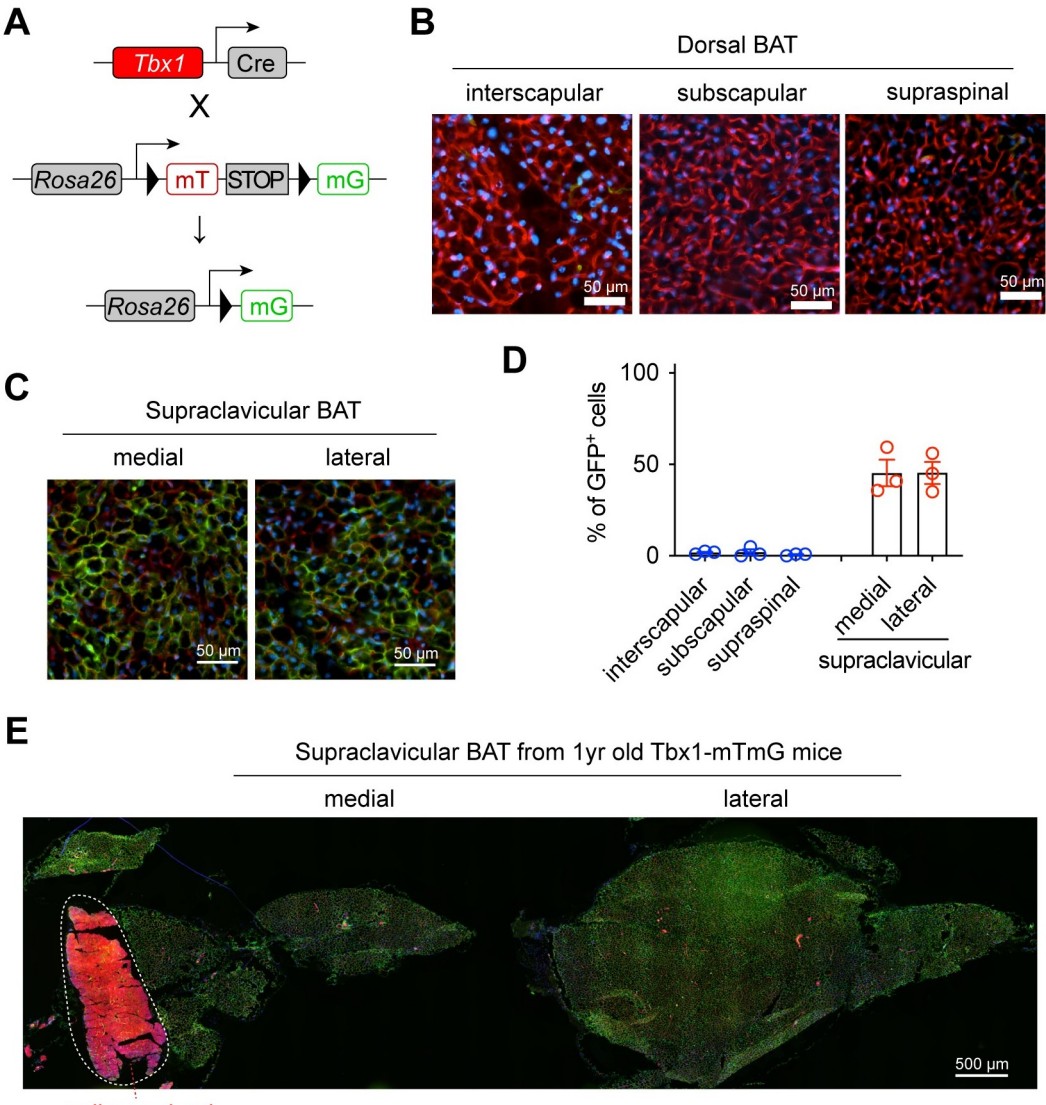

**Fig 4. scBAT is composed of *Tbx1*+ lineage adipocytes.** **(A)** *Tbx1*+ lineage cell labeling with the *Tbx1-mTmG* mice. **(B, C)** Fluorescent images of dorsal BAT (B) and scBAT (C) from 6-week-old female *Tbx1-mTmG* mice (green = mG, red = mT, blue = DAPI, scale = 50 μm). **(D)** Frequency of GFP+ cells within each BAT depot (*n* = 3). Data are presented as mean ± SEM. **(E)** Fluorescent images of scBAT from a 1-year-old *Tbx1-mTmG* mouse (green = mG, red = mT, blue = DAPI, scale = 500 μm). Salivary gland (epithelial lineage) was labeled by mT. Data are presented as mean ± SEM. Extended data are listed in S1 Data. BAT, brown adipose tissue; scBAT, supraclavicular brown adipose tissue.

(Fig 5O) or inguinal WAT (S3F Fig). Nonetheless, male *Pparg*^*f/f*^ and *Pparg*^*ΔTbx1*^ mice had similar *Ucp1* expression and tolerance to cold challenge (Fig 5P), indicating that male mice rely less on scBAT for thermogenesis.

## scBAT paucity does not exacerbate diet-induced obesity

In humans, scBAT prevalence is negatively correlated with BMI and cardiometabolic dysfunction. We thus went on to test if scBAT paucity renders mice more susceptible to high-fat diet (HFD)-induced obesity and complications. We subjected both female and male *Pparg*^*ΔTbx1*^

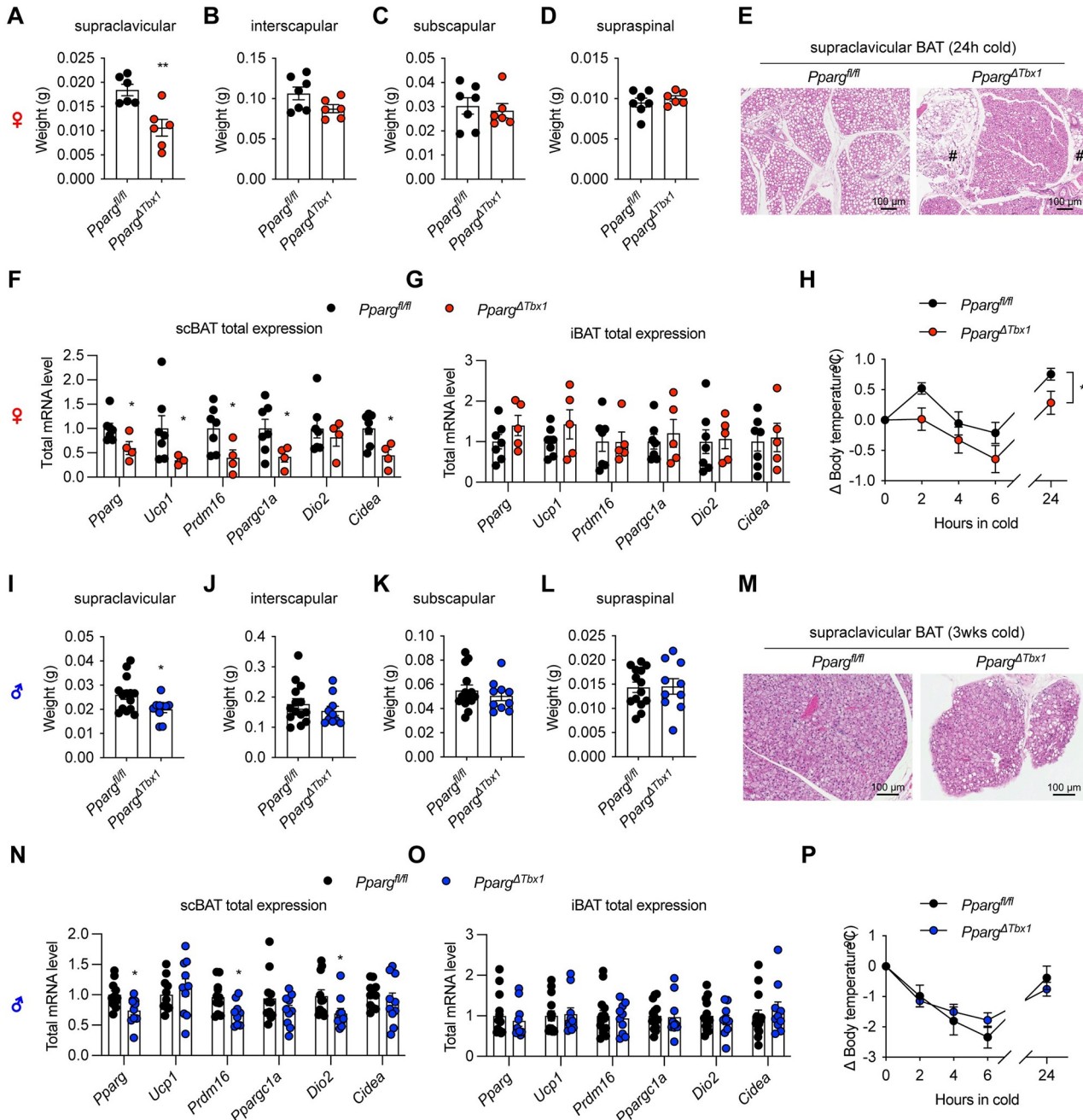

**Fig 5. PPARγ deficiency in the *Tbx1*⁺ lineage impairs scBAT function.** (**A–D**) Weights of supraclavicular (A), interscapular (B), subscapular (C), and supraspinal (D) BAT depots from 4-month-old *Pparg^{f/f}* (*n* = 7) and *Pparg^{ΔTbx1}* (*n* = 6) female mice. (**E**) Representative HE staining of scBAT from *Pparg^{f/f}* and *Pparg^{ΔTbx1}* female mice (scale = 100 μm). # indicates areas of adipose whitening and tissue degeneration. (**F, G**) Thermogenic gene expression in scBAT (F) and iBAT (G) of female mice was determined by RT-qPCR and adjusted by total tissue RNA to calculate the relative total transcript levels. (**H**) Changes in core body temperature of *Pparg^{f/f}* (*n* = 9) and *Pparg^{ΔTbx1}* (*n* = 7) female mice during cold challenge in 4°C. (**I–L**) Weights of supraclavicular (I), interscapular (J), subscapular (K), and supraspinal (L) BAT depots from 4-month-old *Pparg^{f/f}* (*n* = 14) and *Pparg^{ΔTbx1}* (*n* = 10) male mice after 3 weeks of cold challenge. (**M**) Representative HE staining of scBAT from *Pparg^{f/f}* and *Pparg^{ΔTbx1}* male mice (scale = 100 μm). (**N, O**) Thermogenic gene expression in scBAT (N) and iBAT (O) of male mice was determined by RT-qPCR and adjusted by total tissue RNA to calculate the relative total transcript levels. (**P**) Changes in core body temperature of *Pparg^{f/f}* (*n* = 6) and *Pparg^{ΔTbx1}* (*n* = 17) male mice during cold challenge in 4°C. Data are presented as mean ± SEM.*, $p < 0.05$; **, $p < 0.01$; and ***, $p < 0.001$ by unpaired Student's *t* test or two-way ANOVA (H). Extended data are listed in S1 Data. BAT, brown adipose tissue; iBAT, interscapular brown adipose tissue; PPARγ, peroxisome proliferator-activated receptor gamma; scBAT, supraclavicular brown adipose tissue.

mice and their littermate controls to HFD feeding; however, no difference in weight gain was observed between genotypes (Fig 6A and 6B). $Pparg^{\Delta Tbx1}$ mice also had similar tolerance to glucose (Fig 6C), suggesting scBAT paucity in mice is not sufficient to cause metabolic dysfunction.

Obesity induces systemic and hepatic inflammation, and BAT recruitment and UCP1 activation by cold have been suggested to participate in resolving systemic and hepatic inflammation [27,28]. To examine if scBAT is involved in inflammation resolution, we subjected HFD-fed $Pparg^{\Delta Tbx1}$ mice to lipopolysaccharide (LPS) injection to generate acute endotoxemia [29]. Notably, higher serum levels of LPS were observed in $Pparg^{\Delta Tbx1}$ mice compared to controls (Fig 6D), indicative of possible role of scBAT in neutralizing LPS. As expected, $Ucp1$ and $Adipoq$ gene expression was down-regulated in scBAT of HFD-fed, LPS-treated $Pparg^{\Delta Tbx1}$ mice (Fig 6E and 6F). However, we did not observe any significant changes in the expression of inflammatory genes such as $Il1b$, $Il6$, $Tnfa$, and $Ifng$ in either scBAT (Fig 6G) or liver (Fig 6H). We speculate that, because of the intact dorsal BAT depots in these animals, specific paucity of the smaller scBAT would not predispose animals to HFD-induced metabolic dysfunction and inflammation.

## PRMD16 drives the thermogenic programing of scBAT

Next, we investigated the molecular determinants of scBAT development and function. Because of the similar thermogenic activity and regulation between scBAT and iBAT [25,26], we postulated that the transcriptional regulatory circuits for iBAT will control scBAT differentiation and/or activity [30,31]. PRDM16 dictates the brown adipogenic switch of myogenic progenitors [15] and is required for WAT browning and the maintenance of brown adipocyte identify [32,33]. In $Pparg^{\Delta Tbx1}$ mice, $Prdm16$ gene expression was down-regulated in scBAT (Fig 5F and 5N). To investigate the role of PRDM16 in scBAT, we generated $Tbx1^{Cre}$-mediated PRDM16 knockout ($Prdm16^{\Delta Tbx1}$) mice. While we did not observe weight changes in tissues including dorsal BAT depots, scBAT, WAT, and quadricep muscles in female $Prdm16^{\Delta Tbx1}$ mice (Fig 7A), these animals showed more body temperature loss during the cold tolerance test (Fig 7B). HE histology demonstrated adipose whitening when PRDM16 is absent (Fig 7C). RT-qPCR analysis revealed profound down-regulation of thermogenic/adipogenic genes including $Ucp1$, $Pparg$, $Dio2$, $Cidea$, $Pparg1c$, and $Adipoq$ specifically in scBAT, but not iBAT of female $Prdm16^{\Delta Tbx1}$ mice (Fig 7D). Similar findings were observed in male $Prdm16^{\Delta Tbx1}$ mice. Compared to littermate controls, male $Prdm16^{\Delta Tbx1}$ mice were less cold tolerant (Fig 7E), despite of having similar weight of scBAT and other depots (Fig 7F). These data suggest that PRDM16 is critical for the thermogenic function of scBAT in adult mice.

## *TBX1* marks human deep neck BAT

Finally, we determined the expression of *TBX1* gene in human deep neck BAT. Since iBAT was absence in adult humans, subcutaneous neck WAT from healthy donors was obtained as controls [34]. RT-qPCR revealed a much higher levels of *TBX1* expression in total lysates from deep neck BAT than subcutaneous neck WAT (Fig 8A). Adipogenic differentiation of preadipocytes isolated from both subcutaneous WAT and deep neck BAT induced *TBX1* expression (Fig 8B). Nonetheless, a consistent higher expression of *TBX1* could be observed in preadipocytes and adipocytes derived from deep neck BAT, compared to those from subcutaneous neck WAT (Fig 8B). Mimicking adrenergic stimulation, the cell-permeable dibutyryl-cAMP stimulates nutrition uptake and oxygen consumption of human brown adipocytes [35]. However, cAMP did not change *TBX1* expression (Fig 8C), indicating that *TBX1* might not be a

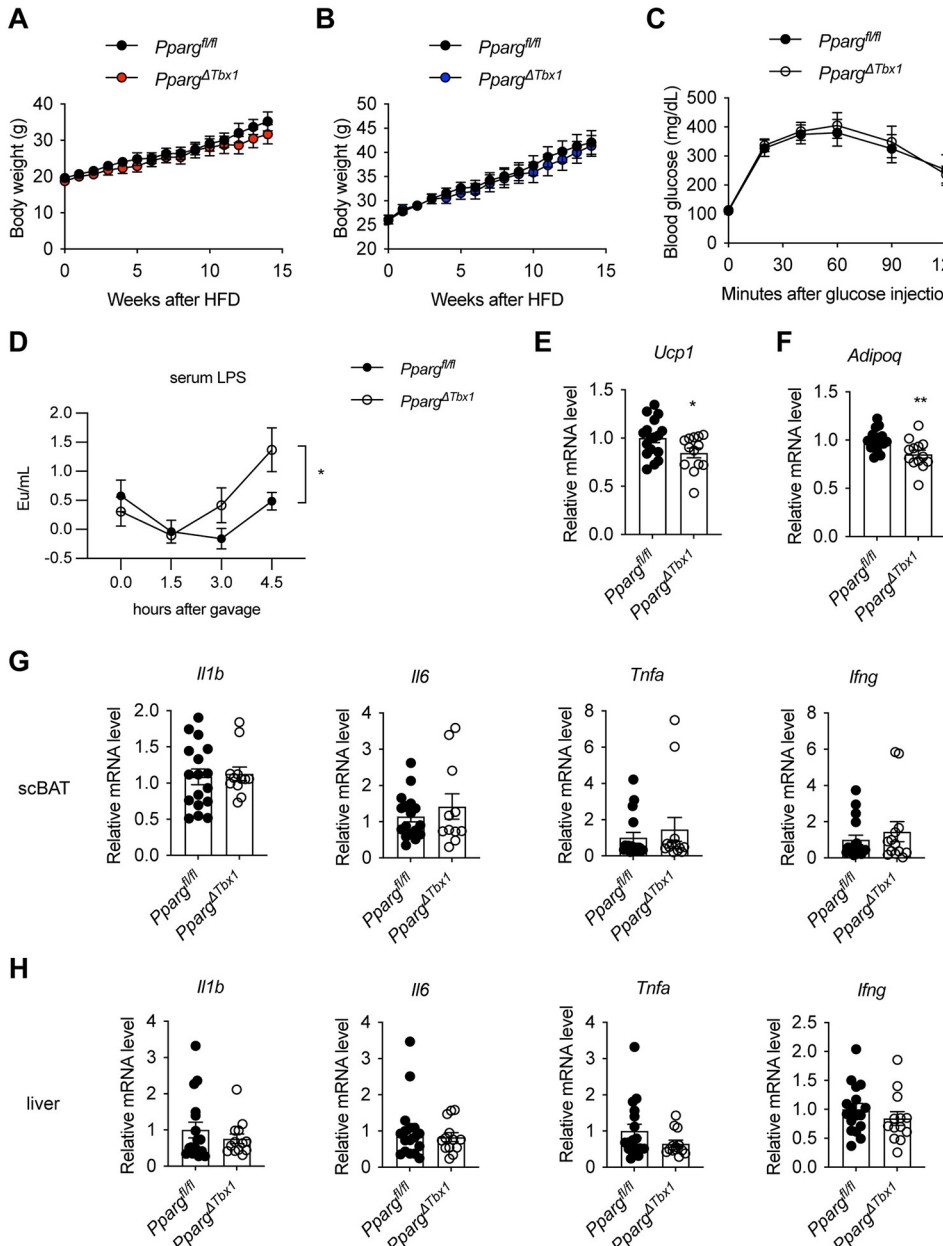

**Fig 6. Characterization of HFD-fed *Pparg^{ΔTbx1}* mice.** (**A, B**) Body weight of female (A, *n* = 6–7) and male (B, *n* = 10–12) *Pparg^{f/f}* and *Pparg^{ΔTbx1}* mice after HFD feeding. (**C**) Glucose tolerant test of HFD-fed *Pparg^{f/f}* (*n* = 8) and *Pparg^{ΔTbx1}* (*n* = 10) mice (mixed sexes). (**D**) HFD-fed *Pparg^{f/f}* (*n* = 10) and *Pparg^{ΔTbx1}* (*n* = 10) male mice were fasted overnight, orally gavaged with 4 µg LPS (in 200 µl olive oil), and sera were collected at indicated time for LPS measurement. (**E–H**) HFD-fed *Pparg^{f/f}* (10 males, 7 females) and *Pparg^{ΔTbx1}* (8 males, 5 females) mice were fasted and gavaged with 50 µl of 0.1 µg/µl LPS in water per 20 g of body weight. scBAT (E–G) and liver (H) were collected 3 h later for RT-qPCR quantification the expression of *Ucp1* (E), *Adipoq* (F), and inflammatory genes (G, H). Data are presented as mean ± SEM. *, *p* < 0.05; **, *p* < 0.01 by unpaired Student's *t* test (E, F) or two-way ANOVA (D). Extended data are listed in S1 Data. HFD, high-fat diet; LPS, lipopolysaccharide; scBAT, supraclavicular brown adipose tissue.

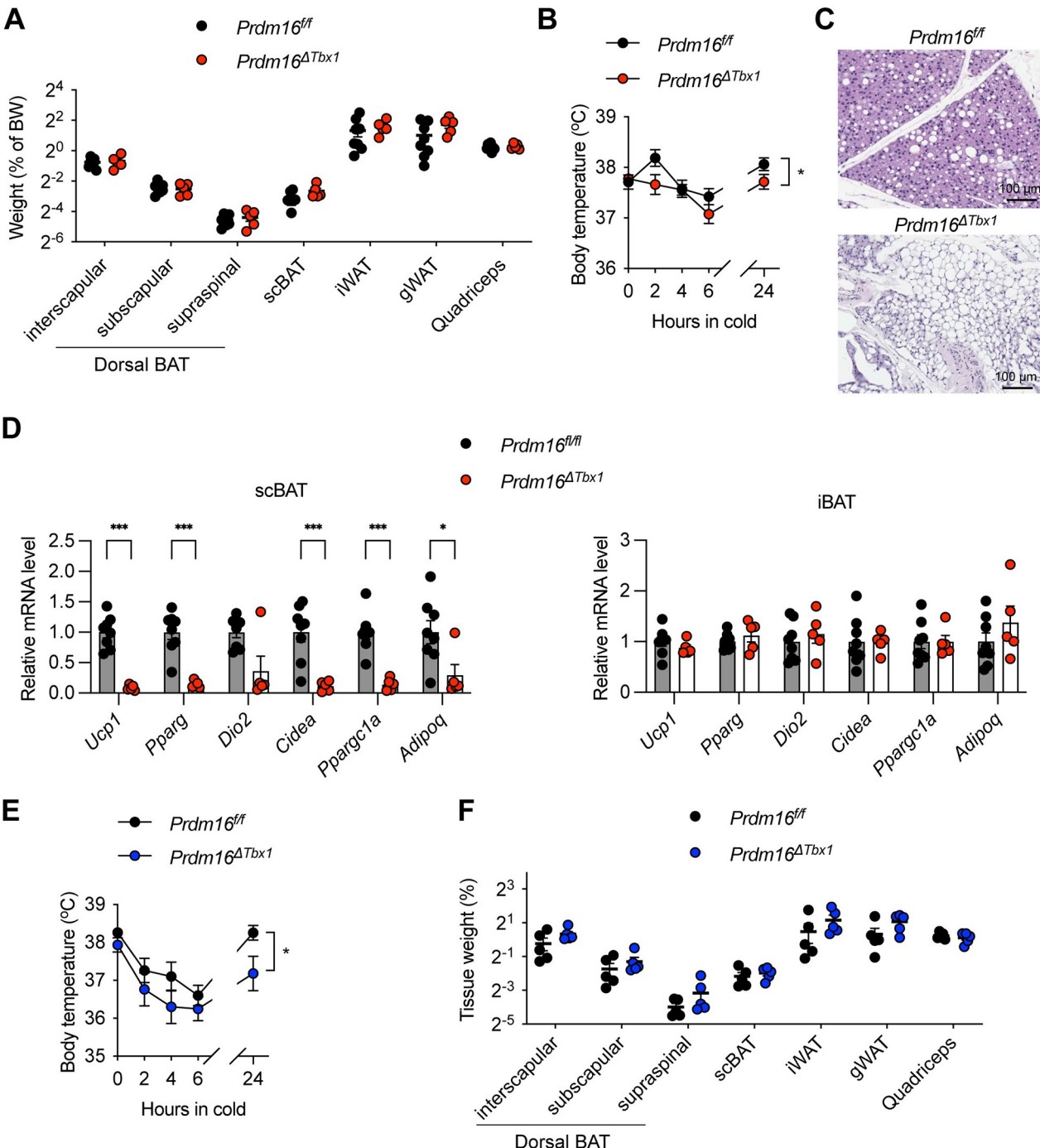

**Fig 7. scBAT dysfunction in *Prdm16^ΔTbx1* mice.** (**A**) Weight of indicated tissues as a percentage of body weight in *Prdm16^f/f* (*n* = 8) and *Prdm16^ΔTbx1* (*n* = 5) female mice. (**B**) Core body temperature of *Prdm16^f/f* (*n* = 8) and *Prdm16^ΔTbx1* (*n* = 5) female mice during cold challenge in 4°C. (**C**) Representative HE staining of scBAT from *Prdm16^f/f* and *Prdm16^ΔTbx1* female mice (scale = 100 μm). (**D**) RT-qPCR measurements of gene expression in scBAT and iBAT of *Prdm16^f/f* (*n* = 8) and *Prdm16^ΔTbx1* (*n* = 5) female mice. (**E**) Core body temperature of *Prdm16^f/f* (*n* = 5) and *Prdm16^ΔTbx1* (*n* = 6) male mice during cold challenge in 4°C. (**F**) Weight of indicated tissues as a percentage of body weight in *Prdm16^f/f* (*n* = 5) and *Prdm16^ΔTbx1* (*n* = 6) male mice after 3 weeks of cold challenge. Data are presented as mean ± SEM. *, $p < 0.05$; **, $p < 0.01$ by unpaired Student's *t* test (D) or two-way ANOVA (B, E). Extended data are listed in S1 Data. iBAT, interscapular brown adipose tissue; scBAT, supraclavicular brown adipose tissue.

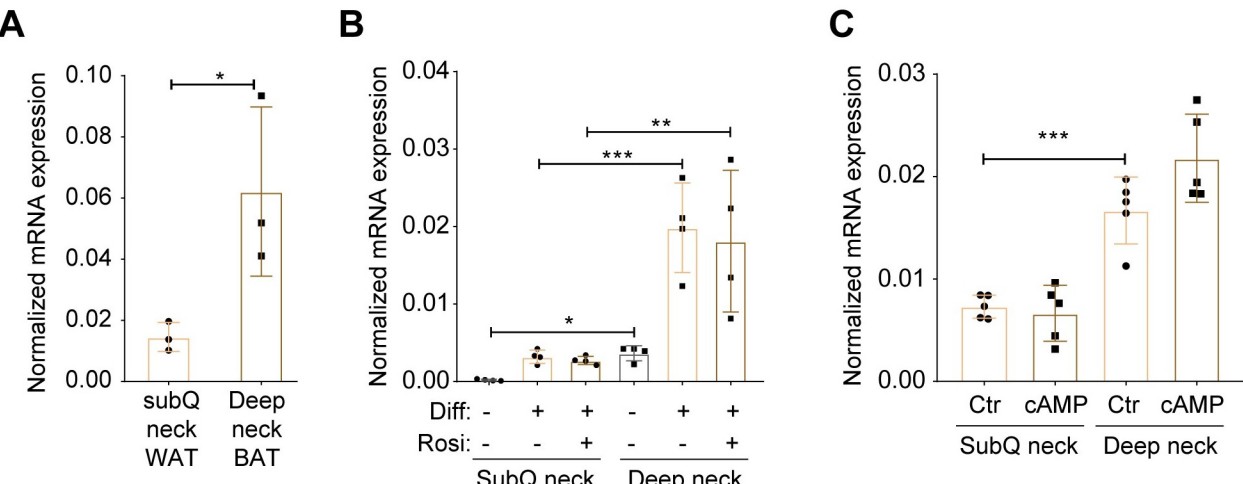

**Fig 8.** ***TBX1* gene expression in human deep neck BAT.** (**A**) Taqman RT-qPCR of *TBX1* gene expression, normalized to *GAPDH*, in subcutaneous (subQ) neck WAT and deep neck BAT from 3 donors. (**B**) SubQ neck WAT and deep neck BAT preadipocytes from 4 donors were differentiated to adipocytes in the presence or absence of rosiglitazone for 14 days. *TBX1* gene expression normalized to *GAPDH* was determined using Taqman probes. (**C**) Differentiated adipocytes from SubQ WAT and deep neck BAT (*n* = 4 donors) were treated with or without 500 μm dibutyril-cAMP for 10 h. *TBX1* gene expression was normalized to *GAPDH*. Data are presented as mean ± SD. *, $p < 0.05$; **, $p < 0.01$ by paired Student's *t* test (A) or two-way ANOVA (B, C). Extended data are listed in S1 Data. BAT, brown adipose tissue.

cold-inducible early gene that contributes to thermogenesis. Instead, based on our mouse data, *Tbx1* is a location marker specifically for scBAT during development.

## Discussion

Obesity is a major risk factor for many diseases, including type 2 diabetes, cardiovascular disease, and some types of cancers. BAT burns fat and dissipates chemical energy as heat; therefore, activating BAT might be a strategy to combat obesity and related metabolic disorders. Since the re-discovery of scBAT in adult humans, many efforts have been devoted in the field to find ways to convert white fat into brown or beige fat. However, almost all the current preclinical studies investigate either iBAT or subcutaneous inguinal WAT in rodents, with assumptions that these depots possess similar developmental, functional, and regulatory mechanisms as the predominant scBAT depot in human adults. In fact, many depot-, location-, and species-specific characteristics of thermogenic adipocytes have been described [4]. For example, human iBAT rapidly atrophies till undetectable in adults [2,3]. However, mice iBAT shows little involution or atrophy. Whitened, hypertrophic iBAT adipocytes in aged or "humanized" mice remain as the progeny of *Myf5*[+] progenitors [17] and can be recruited by cold and sympathetic activation [36–41]. In mice, inguinal WAT adipocytes are derived from *Prrx1*[+] progenitors in the somatic lateral plate mesoderm [42,43]. Inguinal WAT browning is driven by de novo beige adipogenesis [44,45], activation of "dormant" beige cells [46], and "transdifferentiation" of white adipocytes [47,48]. Like that in mice, subcutaneous WAT in humans can undergoes thermogenic activation [49–52]. However, visceral WAT in humans seems to have higher thermogenic capacity than subcutaneous WAT [53].

These depot- and species-specific findings highlight the urgency and importance to understand scBAT development and regulation. Our study here provides the first evidence that scBAT adipocytes do not share the same embryonic origins as iBAT fat cells. Somatic *Pax3*[+]/ *Myf5*[+] myoprogenitors give rise to dorsal-anterior-located iBAT, as well as nearby WAT and

trunk muscles. In contrast, *Tbx1*⁺ myoprogenitors from the CPM give rise to scBAT adipocytes, in addition to many muscle groups in the head and neck. The *Tbx1* gene has been suggested as a marker for beige adipocytes [54]. However, β3 adrenergic agonism does not induce the expression of *TBX1* in human (Fig 8C) or *Tbx1* in mouse [55]. Mouse *Tbx1* was also expressed by inguinal white adipocytes under warm conditions [39], suggesting that *Tbx1* is not a bona fide beige marker, but reflects the anatomical location of the inguinal depot. Due to its high expression in inguinal WAT, we speculate, even though not directly examined here, that inguinal fat cells will be labeled in the *Tbx1-mTmG* mice. However, we would not be able to distinguish between their embryonic lineage origin versus postnatal expression of *Tbx1*, because of the constitutive Cre expression in *Tbx1*^*Cre* mice. Recently, adipose expression of TBX1 was shown to be necessary for UCP1 expression and insulin sensitivity of subcutaneous WAT [56]. Future investigations using inducible *Tbx1*-driven Cre models and depot-specific deletion of TBX1 are required to determine the temporospatial function of TBX1. Nonetheless, the shared expression of TBX1 between scBAT and inguinal WAT also provides additional justification to study WAT browning in contributing to metabolic health.

In lineage mapping experiments, we noticed that only about half of scBAT adipocytes are labeled by *Tbx1*^*Cre*. This could be a result of low Cre expression or insufficient recombination (S3A Fig), supported by the uniform distribution and no cluster formation of mG⁺ adipocytes (Fig 4E). Nonetheless, it is also possible that scBAT adipocytes have multiple developmental origins and *Tbx1* only labels a portion of CPM myoprogenitors. It would be interesting in the future to test whether other CPM markers such as *Ptx2* and *Islet1* [22–24] trace all or some of the scBAT adipocytes. Although controlling myogenesis at different locations, both *Tbx1* and *Myf5* elicit a transcriptional cascade including *Myod* and *Myogenin* [57]. *Myod* also induces glycolytic beige adipocytes in the absence of β-adrenergic receptor signaling [58]. However, *Myod* does not label any iBAT brown adipocytes [17]. It stresses that the bifurcation between brown adipogenesis and myogenesis happens upstream of *Myod*. The PRDM16-C/EBPβ-EHMT1 transcriptional complex specifies the iBAT brown adipocyte fate from *Myf5*⁺ progenitors and the PRDM16-PPARγ-PGC-1α complex drives the complete brown fat differentiation [59,60]. Here, we have showed that both PPARγ and PRDM16 are also important for scBAT development and function. *Tbx1*^*Cre*-mediated deletion of either PPARγ or PRDM16 leads to scBAT dysfunction, evidenced by reduced *Ucp1* expression and cold intolerance. PPARγ deficiency cause early developmental paucity of scBAT, but loss of PRDM16 only causes thermogenic reduction in adult animals. This is consistent with earlier findings based on *Myf5*^*Cre*-mediated PRDM16 KO in iBAT [33], which collectively demonstrate that PRDM16 in myoprogenitors is required for brown adipocyte identity maintenance during aging. While current evidence suggests shared regulatory mechanisms for the recruitment and activation of iBAT and scBAT [25,26], it warrants further investigations to identify depot-specific regulations and functions of BAT, in addition to their distinct developmental origins. In adult BAT, progenitors that are marked by genes like *Pdgfra* and *Trpv1* have been reported to contribute to cold-induced BAT recruitment and tissue homeostasis [47,61,62]. While not the scope of our current research, future endeavors are needed to test if embryonic *Tbx1*+ myoprogenitors give rise to all or only some populations of adult BAT progenitors.

While the absolute mass and thermogenic capacity of human BAT are difficult to quantify, it is well accepted that BAT prevalence declines as a function of age [1,4]. Thus, many have questioned the physiological relevance of BAT in thermogenesis and body temperature control in adult humans, particularly the elderly. The thermogenic contribution of human BAT might be limited; however, it is without doubt that the presence of scBAT is independently correlated with lower incidences of obesity, type 2 diabetes, dyslipidemia, hypertension, and heart failure [14]. It is thus tempting to hypothesize that iBAT is a critical thermogenic organ for infants,

while scBAT in adults primarily modulates systemic metabolism. Anatomically, scBAT sits adjacent to the jugular vein and subclavian vein, where the lymphatic vessels empty lymph collected from the intestine into the venous system. We hypothesized that this unique anatomical location of scBAT may enable its ability to sample and regulate lymphatic fluids, such as bacterial endotoxins therein. While we did see higher serum LPS levels in $Pparg^{\Delta Tbx1}$ mice with scBAT paucity (Fig 6D), glucose metabolism and inflammation resolution seemed to be unaffected. The lack of metabolic dysfunction and endotoxemia in $Pparg^{\Delta Tbx1}$ mice could be due to the partial atrophy (approximately 50%) of scBAT and functional compensation from the intact iBAT. Future $Tbx1^{Cre}$-mediated brown adipocyte ablation and simultaneous removal of iBAT could address these issues.

In summary, the identification of $Tbx1^{+}$ lineage cells as progenitors of scBAT brown adipocytes reveals location-specific myoprogenitors for different BAT depots in rodents and possibly humans. This knowledge can be leveraged to develop new models in order to discover depot-specific BAT functions. Not only should we cease to state that "brown adipocytes are derived from a $Myf5$-expressing lineage," but also study more the $Tbx1^{+}$ lineage-derived brown adipocytes due to their human relevance.

## Methods

### Animals

All animal experiments were approved by the institutional animal care and use committee (IACUC) of the University of Minnesota (protocol #: 2112-39682A) and adhered to the NIH Guide for the Care and Use of Laboratory Animals. All the mice were group-housed in light/dark cycle—(6 AM to 8 PM light), temperature—(21.5 ± 1.5°C), and humidity-controlled (30% to 70%) room, and had free access to water and regular chow (Teklad #2018), or 60% HFD (Research Diet #D12492) as indicated. $Tbx1^{Cre}$ (MGI:3757964) was a kind gift from Dr. Antonio Baldini [63]. $Myf5^{Cre}$ (Jax #007893), $Pax3^{Cre}$ (Jax #005549), $Pparg^{f/f}$ (Jax #004584), and $Rosa26^{LSL-mT/mG}$ (Jax #007676) mice were from Jackson Lab.

To confirm $Pparg$ knockout, RT-PCR was conducted using primers (5′-GTCACGTTCT-GACAGGACTGTGTGAC-3′) and (5′-TATCACTGGAGATCTCCGCCAACAGC-3′) encompassing exons A1 and 4 of the $Pparg1$ gene, which differentiate the full-length (700 bp) and recombined (300 bp) transcripts [64]. PCR was performed on a Bio-Rad C1000 Thermal Cycler using a touchdown program: 10 cycles of 94°C for 20 s, 65°C (−0.5°C/cycle) for 15 s, and 68°C for 15 s, followed by 28 cycles of 94°C for 15 s, 58°C for 15 s, and 72°C for 15 s.

For cold treatment, mice were housed in a temperature-controlled room (4°C) with free access to water and food. Core body temperature was measured using an electronic thermometer with anal probe.

For glucose tolerance test, mice were fasted overnight for 16 h and intraperitoneally injected with 1.5 g/kg body weight of glucose. Blood glucose levels were measured using a Bayer Contour Glucometer at indicated time point after injection.

For LPS treatment, mice were fasted overnight when indicated and orally gavaged with LPS from Escherichia coli O111:B4 (Sigma, L3024) in water or olive oil. Tail blood was collected for LPS measurement using a Pierce chromogenic endotoxin quantification kit (Thermo Fisher, A39552).

### Human adipocytes

Tissue collection was approved by the Medical Research Council of Hungary (20571-2/2017/EKU) followed by the EU Member States' Directive 2004/23/EC on presumed consent practice for tissue collection. All experiments were carried out in accordance with the guidelines of the

Helsinki Declaration. Written informed consent was obtained from all participants before the surgical procedure. During thyroid surgeries, a pair of deep neck BAT and subcutaneous WAT samples was obtained to rule out inter-individual variations. Patients with known diabetes, malignant tumor, or with abnormal thyroid hormone levels at the time of surgery were excluded. Tissue specimens were either homogenized in Trizol or digested in phosphate buffered saline (PBS) with 120 U/ml collagenase (Sigma, C1639) to obtain stromal vascular fraction (SVF). Floating cells were washed away with PBS after 3 days of isolation and the remaining cells were cultured [34]. Human primary adipocytes were differentiated from SVF of adipose tissue containing preadipocytes according either to a regular adipogenic protocol or in the presence of long-term rosiglitazone effect resulting in higher browning capacity of the adipocytes. Where indicated, adipocytes were treated with a single bolus of 500 μm dibutyryl-cAMP (Sigma, D0627) for 10 h to mimic in vivo cold-induced thermogenesis [35]. Then, SVF cells or adipocytes were homogenized using Trizol.

## Histology

Brown adipose tissues were harvested and fixed in 10% formalin overnight with gentle shaking, then kept in 4°C for further experiments. For hematoxylin and eosin staining, BAT embedding, sectioning, staining was conducted at the Comparative Pathology Shared Resource of the University of Minnesota. For the fluorescent imaging, the fixed BAT was embedding with OCT (Tissue-Tek #4583) then sliced into 10 μm slides. Following PBS washing, the sections were mounted using VECTASHIELD Antifade Mounting Medium with DAPI and visualized using Nikon Ni-E or Keyence microscope system. The numbers of Tomato[+] and GFP[+] cells were counted to calculate the percentage of GFP[+] cells.

## RT-qPCR

After weight measurement, BAT tissues were homogenized in Trizol (Thermo Scientific) for RNA isolation, following the manufacturer's protocol. RNA concentrations were measured with a NanoDrop spectrophotometer. Reverse transcription was performed with the iScript cDNA Synthesis Kit. Real-time RT-PCR was conducted using iTaq Universal SYBR Green Supermix and gene-specific primers on a Bio-Rad C1000 Thermal Cycler. Relative expression was normalized to the house keeping *Rplp0* gene. When indicated, total mRNA amounts were calculated based on relative *Ucp1* mRNA levels and total amounts of RNA isolated from specific depots [65]. For human WAT, BAT, and adipocyte lysates, RT-PCR was conducted using validated TaqMan assays (Thermo Fisher, Hs00271949_m1 for *TBX1*, and Hs99999905_m1 for *GAPDH*). Gene expression values were calculated by the comparative threshold cycle (Ct) method. ΔCt represents the Ct of the target minus that of *GAPDH*. Normalized gene expression levels equal 2−ΔCt [35]. Samples with poor RNA quality or artifact bias, determined by housing keeping gene expression and melting curve, were removed from analysis.

## Western blot

Tissues were collected quickly after sacrificing mice and homogenized immediately in RIPA lysis buffer (50 mM Tris–HCl (pH 7.4), 1% Nonidet P-40, 0.25% Na-deoxycholate, 150 mM NaCl, 1 mM EDTA, and protease inhibitors) on ice-water bath. Protein concentration was measured using a BCA protein assay kit (Thermo Fisher). Protein samples were separated by SDS-PAGE and western blots were performed with following antibodies: UCP1 (Abcam, #ab209483), PPARγ (Cell Signaling Techonology, #2443), and Tubulin (Santa cruz, #SC-8035).

## Quantification and statistical analysis

Results are shown as mean ± SEM or ± SD. *N* values (biological replicates) and statistical analysis methods are described in figure legends. The statistical comparisons were carried out using two-tailed Student's *t* test and one-way or two-way ANOVA with indicated post hoc tests with Prism (Graphpad). Differences were considered significant when $p < 0.05$. *, $p < 0.05$; **, $p < 0.01$; ***, $p < 0.001$.

## Supporting information

**S1 Fig. Validation of *mTmG* reporter mice.** Representative fluorescent images of interscapular (left) and supraclavicular (right) BAT from *Cre*-negative *mTmG* reporter mice (scale = 50 μm).
(TIF)

**S2 Fig. Validation of *Tbx1-mTmG* reporter mice.** Representative fluorescent images of scapular (left) and clavicular (right) skeletal muscles from *Tbx1-mTmG* reporter mice (scale = 50 μm).
(TIF)

**S3 Fig. No changes in body and tissue weight of *Pparg^{ΔTbx1}* mice.** (**A**) Detection of wild type (wt, 700 bp) and mutant (mt, 300 bp) *Pparg* transcripts by RT-PCR. Note the approximately 50% recombination of the *Pparg* gene only in scBAT of *Pparg^{ΔTbx1}* mice. (**B, C**) Body weight (B) and tissue weight (C) of 4-month-old *Pparg^{f/f}* ($n = 7$) and *Pparg^{ΔTbx1}* ($n = 6$) female mice. (**D, E**) Body weight (D) and tissue weight (E) of 4-month-old *Pparg^{f/f}* ($n = 14$) and *Pparg^{ΔTbx1}* ($n = 10$) male mice. (**F**) Thermogenic gene expression in inguinal WAT of male mice was determined by RT-qPCR and adjusted by total tissue RNA to calculate the relative total transcript levels.
(TIF)

**S1 Data. Individual numeric results illustrated in all figures.**
(XLSX)

**S1 Raw. Image. The uncropped and unadjusted blots related to Figs 3C and S3A.**
(PDF)

## Acknowledgments

We thank Dr. Antonio Baldni for kindly providing the *Tbx1^{Cre}* mice and Dr. Bernice Morrow for shipping the mice to us. We thank Dr. Jun Wu and Dr. Rita Perlingeiro for sharing the *Prdm16^{f/f}* and *Pax3^{Cre}* mice, respectively. We thank Dr. Ferenc Győry for the surgical removal of human BAT and WAT biopsies.

## Author Contributions

**Conceptualization:** Hai-Bin Ruan.

**Data curation:** Zan Huang, Chenxin Gu, Zengdi Zhang, Rini Arianti, Aneesh Swaminathan, Kevin Tran, Alex Battist, Endre Kristóf, Hai-Bin Ruan.

**Formal analysis:** Zan Huang, Chenxin Gu, Zengdi Zhang, Rini Arianti, Aneesh Swaminathan, Kevin Tran, Alex Battist, Endre Kristóf, Hai-Bin Ruan.

**Funding acquisition:** Zan Huang, Endre Kristóf, Hai-Bin Ruan.

**Methodology:** Zan Huang.

**Resources:** Hai-Bin Ruan.

**Supervision:** Hai-Bin Ruan.

**Writing – original draft:** Zan Huang, Hai-Bin Ruan.

**Writing – review & editing:** Hai-Bin Ruan.

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
