## [Editor Report · Decision Letter 0]

31 Aug 2023

Dear Dr Ruan, 

Thank you for submitting your manuscript entitled "Supraclavicular brown adipocytes originate from Tbx1+ myoprogenitors" for consideration as a Research Article by PLOS Biology, with accompanying reviews from Review Commons.

Your revised manuscript and response to reviewers has now been evaluated by the PLOS Biology editorial staff as well as by an academic editor with relevant expertise and I am writing to let you know that we would like to send your submission back to the original reviewers for their input on the revision.

However, before we can send your manuscript to the reviewers, we need you to complete your submission by providing the metadata that is required for full assessment. To this end, please login to Editorial Manager where you will find the paper in the 'Submissions Needing Revisions' folder on your homepage. Please click 'Revise Submission' from the Action Links and complete all additional questions in the submission questionnaire.

Once your full submission is complete, your paper will undergo a series of checks in preparation for peer review. After your manuscript has passed the checks it will be sent out for review. To provide the metadata for your submission, please Login to Editorial Manager (https://www.editorialmanager.com/pbiology) within two working days, i.e. by Sep 02 2023 11:59PM.

Kind regards,

Luke

Lucas Smith, Ph.D.

Senior Editor

PLOS Biology

lsmith@plos.org

---

## [Decision Letter · Decision Letter 1]

29 Sep 2023

Dear Dr Ruan,

Thank you for your patience while we considered your revised manuscript "Supraclavicular brown adipocytes originate from Tbx1+ myoprogenitors" for publication as a Research Article at PLOS Biology. This revised version of your manuscript has been evaluated by the PLOS Biology editors, the Academic Editor and by the original reviewers from Review Commons, who are largely satisfied by the changes made in the revision. As a note, due to timing constraints, Reviewer 3 was not available to assess the revision in detail - however, s/he did send informal comments indicating that s/he felt the study merited publication at PLOS Biology. We also asked Reviewer 1 to assess the response to Reviewer 3 in detail. 

Based on the positive reviews, we are likely to accept this manuscript for publication. However, before we can editorially accept your study, we think it would be important for you to address Reviewer 1's last suggestion, and we need you to address a number of editorial and policy related requests, detailed below. 

**IMPORTANT: Please address the following editorial requests: 

1) FINANCIAL DISCLOSURES: In the relevant section of our online system, please update your financial disclosures statement to describe the role of any sponsors or funders in the study design, data collection and analysis, decision to publish, or preparation of the manuscript. If the funders had no role in any of the above, include this sentence at the end of your statement: "The funders had no role in study design, data collection and analysis, decision to publish, or preparation of the manuscript."

2) ETHICS STATEMENT: Please update the ethics statement in your manuscript to include the approval number for the protocol approved by the University of Minnesota IACAC. Please also include the specific national or international regulations/guidelines to which your animal care and use protocol adhered. Please note that institutional or accreditation organization guidelines (such as AAALAC) do not meet this requirement. (ex guidelines that might meet this criteria are teh NIH Guide for the care and use of Laboratory Animals) 

3) BLURB: In the relevant section of our online system, please provide a blurb which (if accepted) will be included in our weekly and monthly Electronic Table of Contents, sent out to readers of PLOS Biology, and may be used to promote your article in social media. The blurb should be about 30-40 words long and is subject to editorial changes. It should, without exaggeration, entice people to read your manuscript. It should not be redundant with the title and should not contain acronyms or abbreviations.

4) BLOT AND GEL REPORTING REQUIREMENTS: We require the original, uncropped and minimally adjusted images supporting all blot and gel results reported in an article's figures or Supporting Information files. We will require these files before a manuscript can be accepted so please prepare and upload them now. Please carefully read our guidelines for how to prepare and upload this data: https://journals.plos.org/plosbiology/s/figures#loc-blot-and-gel-reporting-requirements

>>Please provide the uncropped, and unadjusted blots related to figures 3C and S3A

5) DATA: You may be aware of the PLOS Data Policy, which requires that all data be made available without restriction: http://journals.plos.org/plosbiology/s/data-availability. For more information, please also see this editorial: http://dx.doi.org/10.1371/journal.pbio.1001797

a - Supplementary files (e.g., excel). Please ensure that all data files are uploaded as 'Supporting Information' and are invariably referred to (in the manuscript, figure legends, and the Description field when uploading your files) using the following format verbatim: S1 Data, S2 Data, etc. Multiple panels of a single or even several figures can be included as multiple sheets in one excel file that is saved using exactly the following convention: S1_Data.xlsx (using an underscore).

b - Deposition in a publicly available repository. Please also provide the accession code or a reviewer link so that we may view your data before publication. 

>>>Regardless of the method selected, please ensure that you provide the individual numerical values that underlie the summary data displayed in the following figure panels as they are essential for readers to assess your analysis and to reproduce it:

Fig 1E,H; Fig 2D; Fig 3B,C,E,F; Fig 4D; Fig 5A-D,F-L,N-P; Fig 6 A-H; Fig 7A-B,D-F; Fig 8A-C;

Fig S3B-F;

>>>Please also ensure that figure legends in your manuscript include information on where the underlying data can be found, and ensure your supplemental data file/s has a legend.

>>>Please ensure that your Data Statement in the submission system accurately describes where your data can be found.

6) CODE: Per journal policy, if code was generated to support the conclusions of your manuscript, we require that you make it available without restrictions upon publication. Please ensure that the code is sufficiently well documented and reusable, and that your Data Statement in the Editorial Manager submission system accurately describes where your code can be found.

We expect to receive your revised manuscript within two weeks. 

*Published Peer Review History*

*Press*

Sincerely,

Luke

Lucas Smith, Ph.D.

Senior Editor,

lsmith@plos.org,

PLOS Biology

Reviewer remarks:

Reviewer #1: Summary

The authors have replied to all my concerns, and based on the new version of the manuscript, the quality of the study has been significantly strengthened.

In general terms, the study presents a novel and exciting data, illustrating that supraclavicular brown adipocytes marked by Tbx1+ myoprogenitors and not by Pax3+/Myf5+. The functional tests elucidate the significance of these labeled cells. The study does not yet provide a mechanistic exploration. For example, how Tbx-1 regulates scBAT but not iBAT development and through which signaling pathway remains undetermined. Considering human brown adipocytes are mainly found in the supraclavicular region, the discovery still has significant implication.

Minor comments: 

Since all the Cre lines utilized in this study are constitutive Cre and not inducible Cre, I suggest altering the term "genetic fate mapping" to "cell marking tools" to reflect this more accurately. Genetic fate mapping often involves the use of inducible Cre systems to track cell lineages at specific developmental stages.

Reviewer #2: The authors have thoroughly addressed all of my critiques. The revised manuscript has been significantly improved, making it more than suitable for publication.

---

## [Editor Report · Decision Letter 2]

20 Oct 2023

Dear Dr Ruan,

Thank you for your patience while we considered your revised manuscript "Supraclavicular brown adipocytes originate from Tbx1+ myoprogenitors" for publication as a Research Article at PLOS Biology. This revised version of your manuscript has been evaluated by the PLOS Biology editors and the Academic Editor. 

I apologize for our delay in getting back to you with a decision. Overall, we think that the revision looks good and that you have largely addressed the previous editorial and reviewer requests. However, going through the new underlying data, we noticed two issues which need to be resolved before we can accept your study. 

**Please address the following two editorial requests in another revision:

1) WESTERN BLOTS: Thank you for providing the uncropped blots related to the UCP1 and PPARg analysis in Figure 3C and 3F. We noticed that these appear to have been run on different gels, but that you have only provided 1 tubulin blot (which looks like it comes from the same gel as UCP1). This gives the impression that the PPARg as been normalized to a tubulin band run on a different gel, which would be inappropriate.

Can you please update this file (and the figure in your manuscript) to provide the tubulin signal related to PPARg? After discussion with the Academic Editor, we think that it would be essential for the PPARg to be normalized to a loading control run on the same gel. If you need to repeat this experiment to address this request, please let us know and we are happy to extend the deadline for the revision.

2) DATA: Thank you also for providing the underlying data related to your figures as a supplemental file. As we were taking a quick look at this data, we noticed that the data for Fig 5I has 1 fewer samples compared to J-L (which I think come from the same cohort of mice), and in some panels for figure 5 it appears as if samples had been excluded from the underlying data.

Sorry if I missed these details somewhere in the manuscript, but if not already provided, can you please add a brief description in your methods section detailing any criteria used to exclude mice from analyses?

We expect to receive your revised manuscript within two weeks - but again, we would be happy to extend teh deadline for the revision, as needed. 

*Published Peer Review History*

*Press*

Sincerely,

Luke

Lucas Smith, Ph.D.

Senior Editor,

lsmith@plos.org,

PLOS Biology

---

## [Editor Report · Decision Letter 3]

31 Oct 2023

Dear Dr Ruan,

Thank you for the submission of your revised Research Article "Supraclavicular brown adipocytes originate from Tbx1+ myoprogenitors" for publication in PLOS Biology, and thank you for addressing our last editorial requests in this revision. On behalf of my colleagues and the Academic Editor, Marianne E. Bronner, I am pleased to say that we can in principle accept your manuscript for publication, provided you address any remaining formatting and reporting issues. These will be detailed in an email you should receive within 2-3 business days from our colleagues in the journal operations team; no action is required from you until then. Please note that we will not be able to formally accept your manuscript and schedule it for publication until you have completed any requested changes.

PRESS

Sincerely, 

Lucas Smith, Ph.D.

Senior Editor

PLOS Biology

lsmith@plos.org